# Risk Factors for Neonatal/Maternal Morbidity and Mortality in African American Women with Placental Abruption

**DOI:** 10.3390/medicina56040174

**Published:** 2020-04-13

**Authors:** Deena Elkafrawi, Giovanni Sisti, Sarah Araji, Aldo Khoury, Jacob Miller, Brian Rodriguez Echevarria

**Affiliations:** 1Department of Obstetrics and Gynecology, St. Joseph’s Regional Medical Center, Paterson, NJ 07503, USA; saraharaji@gmail.com (S.A.); khoury3@gmail.com (A.K.); jacobmillermd@gmail.com (J.M.); 2Department of Obstetrics and Gynecology, New York Health and Hospitals/Lincoln, Bronx, NY 10451, USA; gsisti83@gmail.com (G.S.); rodrigub20@nychhc.org (B.R.E.); 3Department of Obstetrics and Gynecology, University of Mississippi Medical Center, Jackson, MI 39216, USA

**Keywords:** African American, antepartum, high risk pregnancy, placental abruption

## Abstract

*Background and Objectives*: Risk factors for neonatal/maternal morbidity and mortality in placental abruption have been incompletely studied in the current literature. Most of the research overlooked the African American population as mostly Caucasian populations are selected. We aimed to find which risk factor influence the neonatal and maternal outcome in cases of placental abruption occurring in African American pregnant women in an inner-city urban setting. *Materials and Methods*: We performed a retrospective cohort study at St. Joseph’s Regional Medical Center, NJ United States of America (USA), between 1986 and 1996. Inclusion criteria were African American race, singleton pregnancy with gestational age over 20 weeks and placental abruption. Maternal age, gravidity, parity, gestational age at delivery/occurrence of placental abruption and mode of delivery were collected. Risk factors for placental abruption such as placenta previa, hypertensive disorders of pregnancy, cigarette smoking, crack/cocaine and alcohol use, mechanical trauma, preterm premature rupture of membranes (PPROM), and premature rupture of membranes (PROM) were recorded. Poor neonatal outcome was considered when anyone of the following occurred: 1st and 5th minute Apgar score lower than 7, intrauterine fetal demise (IUFD), perinatal death, and neonatal arterial umbilical cord pH less than 7.15. Poor maternal outcome was considered if any of the following presented at delivery: hemorrhagic shock, disseminated intravascular coagulation (DIC), hysterectomy, postpartum hemorrhage (PPH), maternal intensive care unit (ICU) admission, and maternal death. *Results:* A population of 271 singleton African American pregnant women was included in the study. Lower gestational age at delivery and cesarean section were statistically significantly correlated with poor neonatal outcomes (*p =* 0.018; *p* < 0.001; *p =* 0.015) in the univariate analysis; only lower gestational age at delivery remained significant in the multivariate analysis (*p =* < 0.001). Crack/cocaine use was statistically significantly associated with poor maternal outcome (*p =* 0.033) in the univariate analysis, while in the multivariate analysis, hemolysis, elevated enzymes, low platelet (HELLP) syndrome, crack/cocaine use and previous cesarean section resulted significantly associated with poor maternal outcome (*p =* 0.029, *p =* 0.017, *p =* 0.015, *p =* 0.047). PROM was associated with better neonatal outcome in the univariate analysis, and preeclampsia was associated with a better maternal outcome in the multivariate analysis. *Conclusions*: Lower gestational age at delivery is the most important risk factor for poor neonatal outcome in African American women with placental abruption. Poor maternal outcome correlated with HELLP syndrome, crack/cocaine use and previous cesarean section. More research in this understudied population is needed to establish reliable risk factors and coordinate preventive interventions.

## 1. Introduction

Placental abruption is the premature separation of the placenta before birth, after 20 weeks of gestation [1]. In the United States of America (USA), clinically detected placental abruption increased by approximately 25 percent in the past decades and a significant increase was noted amongst African American women [2,3]. Placental abruption is a leading cause of maternal and neonatal morbidity and mortality [4,5]. Maternal complications include hemorrhage, disseminated intravascular coagulation (DIC), and hemorrhagic shock. Placental abruption causing fetal death occurs in approximately 1 of every 420 deliveries [1]; additionally, the majority of pregnancies complicated by abruption result in infants weighing less than 10th percentile for gestational age [6,7,8]. While the exact etiology of placental abruption is not well-defined in the literature, a plethora of associated risk factors have been established [8,9]. The most important risk factor for placental abruption is an abruption in a prior pregnancy [10,11,12,13]. Chronic hypertension is a pertinent risk factor for abruption—it is associated with a fivefold increase in risk, which rises to eightfold with superimposed preeclampsia [14]. Placental abruption is 2.5 times more likely to happen in cigarette smokers than non-smokers [15,16,17,18].

Even if the rate of placental abruption is increasing in African American women, they have been neglected by the majority of the research in the current literature, which mostly includes Caucasian women.

In this retrospective study, we evaluated the correlation between various risk factors of placental abruption and neonatal/maternal morbidity and mortality in African American women in an inner-city urban setting.

We described the risks factors and demographics of our population and then measured the association between risk factors and neonatal and maternal outcome.

## 2. Materials and Methods

We performed a retrospective cohort study of all cases of African American patients with placental abruption at St. Joseph’s Regional Medical Center, NJ (USA), between 1986 and 1996. The Institutional Review Board (IRB) approved this study on March 10, 2017, the approval number is 92019j.

The diagnosis of placental abruption was based on clinical findings of abdominal pain, vaginal bleeding, uterine contractions, fetal distress and vital sign abnormalities. We excluded women with multiple gestation and gestational age less than 20 weeks.

We decided to include every case from 20 weeks because the definition of placental abruption is possible at a gestational age > 20 weeks [19] and the lower gestational age limit for viability is still uncertain, indeed between 20 and 25 weeks there is a gray area with up to 50% of survival rate [20].

Age, gravidity, parity, and gestational age at diagnosis/delivery were collected for all patients. Gestational age was calculated using the last menstrual period and confirmed by an obstetric ultrasound in the first trimester, according to the American College of Obstetrics and Gynecology (ACOG) guidelines [21]. Risk factors associated with placental abruption such as mechanical trauma, preterm premature rupture of membranes (PPROM), premature rupture of membranes (PROM), chronic hypertension, preeclampsia, eclampsia, hemolysis, elevated liver enzymes, low platelet (HELLP) syndrome, placenta previa, mode of delivery, history of previous cesarean section, intrauterine growth retardation (IUGR), gestational diabetes mellitus (GDM), pre-GDM, smoking, alcohol, and crack/cocaine usage were recorded. The use of alcohol and crack/cocaine was documented by urine toxicology exam at admission. We considered as main outcome the occurrence of poor neonatal and maternal outcome. A composite poor neonatal outcome was considered present if any of the following was present: neonatal arterial umbilical cord pH at birth lower than 7.15, intrauterine fetal demise (IUFD), perinatal death, Apgar score lower than 7 at 1st and 5th minute of life. A composite poor maternal outcome was considered present if any of the following was present: hemorrhagic shock, disseminated intrauterine coagulation (DIC), hysterectomy, postpartum hemorrhage (PPH) (defined as estimated blood loss of more than 1000 mL according to the ACOG guidelines [22]), maternal intensive care unit (ICU) admission, maternal death. Statistical analysis was performed with the software SPSS (v. 26 IBM, Chicago, US). Normality of the continuous variables was checked with the Skewness and Kurtosis normality test. All the continuous variables resulted not-normally distributed. The Mann–Whitney test was used to compare patients with poor neonatal/maternal outcomes to patients with good neonatal/maternal outcomes. Chi square test was used for categorical variables. Receiver operating characteristic (ROC) curve analysis was used to estimate the association between continuous variable and the dichotomic outcome and the area under the curve (AUC) was calculated accordingly. The Youden index was used to establish the cut-off value for the continuous variables associated with the dichotomic outcome. A p-value of < 0.05 was considered statistically significant.

## 3. Results

Our initial population included 748 pregnant African American women who had placental abruption; however, 477 were excluded due to incomplete perinatal outcome data. Our final sample size was 271 pregnant women with placental abruption: 211 of 271 were preterm deliveries (gestational age less than 37 weeks) (77.8%) and 60 of 271 were term deliveries (gestational age more than 37 weeks) (22.1%).

### 3.1. Risk Factors

Sixteen of 271 pregnant patients with placental abruption had HELLP syndrome (5.90%), 76 of 271 had preeclampsia (28.0%), 22 of 271 had chronic hypertension (8.11%) and 4 of 271 had eclampsia (1.48%). Eighty of 271 had PROM (29.5%). Four of 271 patients had a mechanical trauma as a cause of placental abruption (1.48%), 40 of 271 had crack/cocaine use as a risk factor (14.7%), 53 of 271 smoked less than 10 cigarettes a day (19.5%), 51 of 271 smoked more than 10 cigarettes a day (18.8%), and 45 of 271 had alcohol use (16.6%). Nine of 271 pregnant women with placental abruption had placenta previa (3.32%).

### 3.2. Fetal Complications and Prognosis

One hundred and seventy one of 271 infants were alive on discharge (63.1%), 37 of 271 were stillborn (13.6%), and 44 of 271 were expired on discharge, (16.2%)—42 of the 44 neonatal deaths occurred in preterm deliveries (95.5%) and 2 of 44 neonatal deaths occurred in term deliveries (4.5%).

### 3.3. Correlation of Risk Factors for Placental Abruption with Neonatal and Maternal Morbidity and Mortality

Cesarean section as mode of delivery was statistically significantly associated with poor neonatal outcome (*p =* 0.018 and *p =* 0.015) in the univariate analysis (Table 1). A lower gestational age was associated with increased neonatal morbidity and mortality via univariate and multivariate analysis (*p* < 0.001 in both) (Table 1). The ROC curve analysis showed a significant increase of neonatal risk below 31 weeks of gestation (AUC 0.743, *p* < 0.001) (Figure 1). Crack/cocaine use was associated with poor maternal outcome via univariate and multivariate analysis (*p =* 0.033 and *p =* 0.015) (Table 2). HELLP syndrome and having a history of previous cesarean delivery were correlated with maternal morbidity and mortality (*p =* 0.017 and *p* = 0.047) using the multivariate analysis (Table 2).

PPROM was associated with a better neonatal outcome in the univariate analysis (*p* = 0.018) and preeclampsia was associated with a better maternal outcome in the multivariate analysis (*p* = 0.029) (Table 2).

## 4. Discussion

Placental abruption is rapidly increasing in the USA with a significant portion of abruptions occurring in African American women [2,3]. Placental abruption is a leading cause of obstetric hemorrhage and perinatal death—20% of all fetal deaths from abruption occur after presenting to a hospital with 30% of those deaths occurring within 2 h of admission [1,23]. In one study approximately 40–60% of placental abruption occur before 37 weeks of gestation; the incidence of placental abruption peaks between 24 and 26 weeks [10,14].

The exact cause of placental abruption is unknown—a prevalent hypothesis indicates that placental abruption is secondary to vascular malformations and fragile vasculature resulting in hematoma formation and ultimately placental separation [10]. Approximately 10% of all preterm births occur because of abruption and are associated with increased rates periventricular leukomalacia and intraventricular hemorrhage [10,24]. Placental abruption leads to a decrease in placental surface area necessary for oxygenation, ultimately leading to increased fetal morbidity and mortality [25]. Risk factors such as hypertensive disorders of pregnancy, cocaine use, alcohol use, trauma, cigarette smoking, prior history of placental abruption, and PPROM have all been associated with placental abruption [1,2,3,4,5,6,7,8,9,10,11,12,13,14,15,16,17]. However, our findings indicate that these established risk factors do not correlate with poor neonatal outcomes in African American women. Our results indicate that gestational age of less than 31 weeks and mode of delivery via cesarean section were the only risk factors statistically significant for increased neonatal morbidity and mortality via univariate analysis. A gestational age lower than 31 weeks was the only risk factor significantly associated with neonatal risk in the multivariate analysis.

Only crack/cocaine use was associated with maternal increased maternal morbidity and mortality in the univariate analysis. HELLP syndrome, history of a previous cesarean delivery and crack/cocaine use were statistically significantly associated with increased maternal morbidity and mortality using the multivariate analysis.

PPROM appears to be protective for the neonatal outcome in the univariate analysis, and preeclampsia appears to be protective for maternal outcome in the multivariate analysis. The clinical value of these latter findings is less obvious, and we speculate that the diagnosis of PPROM/preeclampsia may have helped towards a promptly diagnosis of placental abruption and therefore to a quicker and safer delivery. 

Limitations of our study were its retrospective nature and the time frame, it ended in 1996 so it is not the most current. In addition, we did not record the data regarding a previous occurrence of placental abruption.

A strength of the study is the belonging of the whole cohort of patients to a single center study, making the study group less heterogeneous.

## 5. Conclusions

Low gestational age at delivery is the most important risk factor for poor neonatal outcome in African American women with placental abruption. HELLP syndrome, crack/cocaine use and previous cesarean section correlated with poor maternal outcome. More research in this understudied population is needed to establish a reliable risk factor pattern and coordinate preventive interventions.

## Figures and Tables

**Figure 1 medicina-56-00174-f001:**
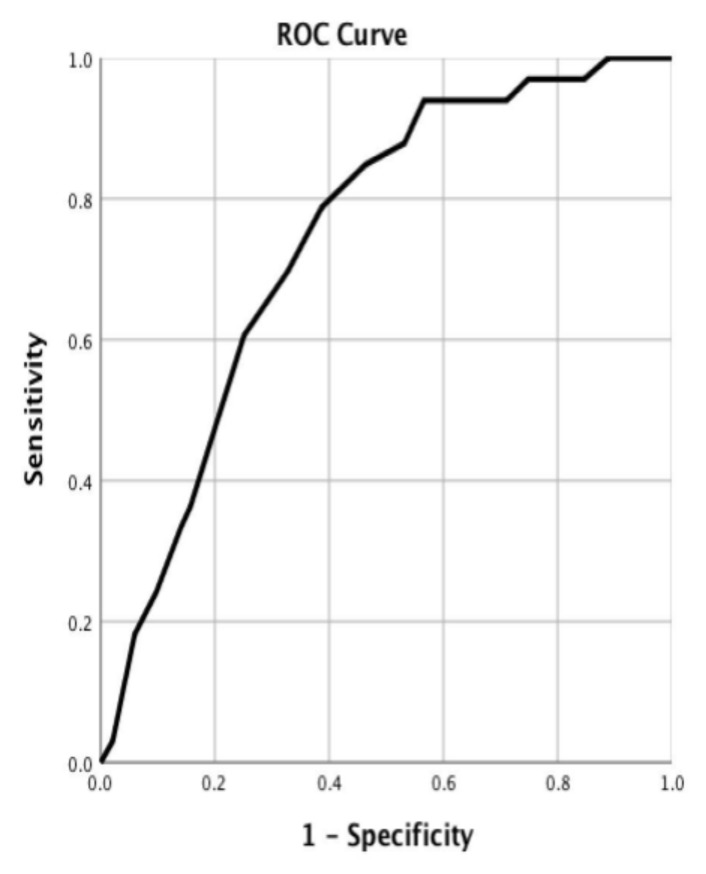
Receiver operating characteristic (ROC) curve of the association between gestational age at delivery/placental abruption and neonatal outcome. Area under the curve (AUC): 0.743; *p* < 0.001.

**Table 1 medicina-56-00174-t001:** Risk factors and their association with neonatal outcome.

Variables	Poor Neonatal Outcome *n* = 237	Good Neonatal Outcome *n* = 35	*p*-Value *	*p*-Value **
Preeclampsia	58/237 (24%)	8/35 (22.8%)	NS	NS
Eclampsia	3/237 (1.2%)	0/35 (0%)	NS	NS
HELLP	11/237 (4.6%)	2/35 (5.7%)	NS	NS
CHTN	14/237 (5.9%)	1/35 (2.8%)	NS	NS
PROM	2/237 (0.8%)	1/35 (2.8%)	NS	NS
PPROM	73/237 (3%)	4/35 (11.4%)	0.018	NS
Mechanical trauma	4/237 (1.6%)	1/35 (2.8%)	NS	NS
Crack/cocaine	38/237 (16%)	6/35 (17.1%)	NS	NS
Smoking	55/237 (23.2%)	4/35 (11.4%)	NS	NS
Alcohol	42/237 (17.7%)	6/35 (17.1%)	NS	NS
Placenta previa	7/237 (2.9%)	2/35 (5.7%)	NS	NS
GDM	9/237 (3.7%)	1/35 (2.8%)	NS	NS
PreGDM	1/237 (0.4%)	0/35 (0%)	NS	NS
Previous c section	40/237 (16.8%)	4/35 (11.4%)	NS	NS
IUGR	5/237 (2.1%)	0/35 (0%)	NS	NS
Gravidity > 1	57/237 (24%)	24/35 (68.5%)	NS	NS
Parity > 0	164/237 (56.5%)	22/35 (62.8%)	NS	NS
Mode of delivery (c section/total)	152/237 (64.1%)	15/35 (42.8%)	0.015	NS
Age (years)	24 (20–29)	25 (21–31.75)	NS	NS
GA at delivery (weeks)	31 (26.8–35)	35.5 (33–38.7)	<0.001	<0.001

Categorical variables are shown as ratio (%); continuous variables are shown as median (25°–75° percentile). * Chi square test for categorical variables and Mann-Whitney test for continuous variables. ** Multivariate logistic regression analysis. Hemolysis, elevated enzymes, low platelet (HELLP), chronic hypertension (CHTN). premature rupture of membranes (PROM). preterm premature rupture of membranes (PPROM), gestational diabetes mellitus (GDM), intrauterine growth retardation (IUGR), cesarean section (c section), gestational age (GA), not significant (NS).

**Table 2 medicina-56-00174-t002:** Risk factors and their association with poor maternal outcome.

Variables	Poor Maternal Outcome *n* = 26	Good Maternal Outcome *n* = 246	*p*-Value *	*p*-Value **
Preeclampsia	4/26 (15.3%)	62/246 (25.2%)	NS	0.029
Eclampsia	0/26 (0%)	3/246 (1.2%)	NS	NS
HELLP	3/26 (11.5%)	10/246 (2.8%)	NS	0.017
CHTN	1/26 (3.8%)	14/246 (5.6%)	NS	NS
PROM	0/26 (0%)	3/246 (1.2%)	NS	NS
PPROM	7/26 (26.9%)	70/246 (28.4%)	NS	NS
Mechanical trauma	0/26 (0%)	5/246 (2%)	NS	NS
Crack/cocaine	8/26 (30.7%)	36/246 (14.6%)	0.033	0.015
Smoking	9/26 (34.6%)	50/246 (20.3%)	NS	NS
Alcohol	7/26 (26.9%)	41/245 (16.7%)	NS	NS
Placenta previa	0/26 (0%)	9/246 (3.6%)	NS	NS
GDM	1/26 (3.8%)	9/246 (3.6%)	NS	NS
PreGDM	0/26 (0%)	1/246 (0.4%)	NS	NS
Previous c section	6/26 (23%)	38/246 (15.4%)	NS	0.047
IUGR	0/26 (0%)	5/246 (2%)	NS	NS
Gravidity > 1	16/26 (61.5%)	188/246 (76.4%)	NS	NS
Parity > 0	16/26 (61.5%)	170/246 (69.1%)	NS	NS
Mode of delivery (c section/total)	15/26 (57.6%)	152/246 (61.7%)	NS	NS
Age (years)	27 (21–30)	23 (20–29)	NS	NS
GA at delivery (weeks)	32 (26–34)	32 (27–36)	NS	NS

Categorical variables are shown as ratio (%); continuous variables are shown as median (25°–75° percentile). * Chi square test. ** Multivariate logistic regression analysis. Hemolysis, elevated enzymes, low platelet (HELLP), chronic hypertension (CHTN), preterm premature rupture of membranes (PPROM), premature rupture of membranes (PROM), gestational diabetes mellitus (GDM), intrauterine growth retardation (IUGR), cesarean section (c section), gestational age (GA), not significant (NS).

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
