# Peer review of "Risk Factors for Neonatal/Maternal Morbidity and Mortality in African American Women with Placental Abruption"

_medicina, 2020, doi:10.3390/medicina56040174_

Round 1

Reviewer 1 Report

There is no approval of the ethics committee to carry out the study. African-American women with placental abruption were studied. If authors wanted to determine racial differences they should have to compare with Caucasian or Hispanic women control groups. Otherwise, there is unethical to select patients on a racial basis.    

The study is a retrospective design and on very old data - almost 25 years old. The study period is from 1986 to 1996. Medical care has improved significantly since then.

A cohort of 748 placental abruption cases was eligible to be included in the study but only 271 were studied i.e. almost 1/3 of the cohort.

 There are some concerns about methodology as well: one of the inclusion criteria is 20 weeks gestation - all we know that up to 23 weeks gestation neonatal viability is exceptional, so evaluation of neonatal outcome is questionable.

The study has a retrospective design and descriptive statistics so the risk factors mean that in women with placental abruption these factors were determined but in fact, the risk factors have to show the probability of the event and weren't studied.

Conclusions: It is well known not only in African-American women that lower gestational age at delivery causes poorer neonatal outcomes than in normal terms and especially after placental abruption. 

Only few references are up to date and the majority is almost 20 years old. Some of them don't meet uniform requirements (15, 23).

Author Response

Reviewer 1 comments and responses by the authors:

  • There is no approval of the ethics committee to carry out the study

The study has IRB approval. IRB number is 92019J

  • African-American women with placental abruption were studied. If authors wanted to determine racial differences they should have to compare with Caucasian or Hispanic women control groups. Otherwise, there is unethical to select patients on a racial basis.
  • Thank you for your observation. African-American women is an understudied part of the population. They have the highest rate of morbidity and mortality but still in the world literature they are overlooked. We decided to focus on this ethnic group to specifically free this group from any racial or ethical bias and find a way to help them. We encourage other research groups to follow our example and we are glad that other researchers are already doing so. Please see reference https://www.ncbi.nlm.nih.gov/pubmed/31553835 for example on NEJM.

  • The study is a retrospective design and on very old data - almost 25 years old. The study period is from 1986 to 1996. Medical care has improved significantly since then.
  • Thank you for your comment. We acknowledge these as limitations of our study in the last lines of the discussion.

  • A cohort of 748 placental abruption cases was eligible to be included in the study but only 271 were studied i.e. almost 1/3 of the cohort.
  • We specified the reason for eliminating such a high number of patients in the result section of the manuscript: “incomplete perinatal outcome data”. Even though this could be seen as a limitation of the current study, it highlights the careful review of each single patient’s chart. In addition, the fact that the whole cohort is from a single center study adds value to the study, we added this as a strength of the study in the discussion.

  • There are some concerns about methodology as well: one of the inclusion criteria is 20 weeks gestation - all we know that up to 23 weeks gestation neonatal viability is exceptional, so evaluation of neonatal outcome is questionable.
  • We decided to include every case from 20 weeks because the definition of placenta abruptio is possibile at a GA > 20 w (see https://www.marchofdimes.org/complications/placental-abruption.aspx# ) and the lower GA limit for viability is still uncertain and between 20 and 25 weeks there is a gray area with up to 50% of survival rate (see https://www.ncbi.nlm.nih.gov/pubmed/31618324 ). We added this explanation in material and methods section so that the reader is aware of these considerations.

  • The study has a retrospective design and descriptive statistics so the risk factors mean that in women with placental abruption these factors were determined but in fact, the risk factors have to show the probability of the event and weren't studied.
  • The nature of the study: observational retrospective can lead us only to establish associations and not causations.

  • Conclusions: It is well known not only in African-American women that lower gestational age at delivery causes poorer neonatal outcomes than in normal terms and especially after placental abruption.
  • We established that this was the only important risk factors, all the other do not matter that much. This will give a useful hint to the clinician challenged by the dramatic event of placental abruptions: he/she will pay more attention to the GA more than any other possible concurrent risk factor, and weight it in his/her clinical decision.

  • Only few references are up to date and the majority is almost 20 years old. Some of them don't meet uniform requirements (15, 23).

Placental abruption is an understudied topic and that is why there are not many recent articles in the literature, therefore adding value to our article. We added 3 references from 2019: [https://www.ncbi.nlm.nih.gov/pubmed/31131918, https://www.ncbi.nlm.nih.gov/pubmed/30910143, https://www.marchofdimes.org/complications/placental-abruption.aspx#

Reviewer 2 Report

I have read this paper with great interest and value the focused approach taken. 

The authors have provided a risk factor and outcome (maternal, neonatal) analysis in a specific cohort of women of african American race who had an abruptio placenta during their pregnancy.

Why is the time window 1986-1996, and not more recent as outcome variables are short term aspects ?

How confident are the authors on e.g. cocaine use ?

As the a priori hypothesis was driven by potential ‘racial’ and/or related socio-economic aspects, how do these data compare to other cohorts of the same or similar time interval and to control African American race cohorts ?

A very relevant portion of cases has been removed during the analysis (748 to 271) because of incomplete perinatal outcome data ?

The last ‘alinea’ of the results section still has to be removed.

Author Response

Reviewer 2 comments and responses by the authors:

  • I have read this paper with great interest and value the focused approach taken.

The authors have provided a risk factor and outcome (maternal, neonatal) analysis in a specific cohort of women of african American race who had an abruptio placenta during their pregnancy.

Why is the time window 1986-1996, and not more recent as outcome variables are short term aspects ?

  • Thank you for your question. The time frame indicated is the one for which we had the most complete data for; since than the management of placental abruption has not changed much so we felt that the data is still interesting for the reader.

  • How confident are the authors on e.g. cocaine use ?
  • The use of illicit drug was carefully documented on the chart and the anamnestic data was confirmed by urine toxicology at admission. We added this information in the maternal and methods section.

  • As the a priori hypothesis was driven by potential ‘racial’ and/or related socio-economic aspects, how do these data compare to other cohorts of the same or similar time interval and to control African American race cohorts ?
  • Our was the first study to use a cohort of African American women with placental abruption to find risk factors for fetal and maternal outcomes. However, in the discussion we wrote “Risk factors such as hypertension disorders of pregnancy, cocaine use, alcohol use, trauma, cigarette smoking, prior history of placental abruption, multi-gestation pregnancy and preterm premature rupture of membranes have all been associated with placental abruption [1-17]. However, our findings indicate that these established risk factors do not correlate with poor neonatal outcomes in African American women.” indicating the novelty of our study compared to other cohorts.

  • A very relevant portion of cases has been removed during the analysis (748 to 271) because of incomplete perinatal outcome data ?
  • We specified the reason for eliminating such a high number of patients in the result section of the manuscript: “incomplete perinatal outcome data”. Even though this could be seen as a limitation of the current study, it highlights the careful review of each single patient’s chart. In addition, the fact that the whole cohort is from a single center study adds value to the study, we added this as a strength of the study in the discussion.

  • The last ‘alinea’ of the results section still has to be removed.
  • The formatting of the text will be done by the editor during the publication process

Round 2

Reviewer 1 Report

Sorry for that, I'm not convinced about the scientific value of your manuscript.

There is almost nothing scientific in this manuscript except retrospective, nonrandomized, unblinded and without a control group, descriptive statistics - that means - overall 0/1 Jadad score for the scientific value of the study. 

Reviewer 2 Report

the limitations have been acknowledged.